  

# Proteomic insights into extracellular matrix dynamics in the intestine of *Labeo rohita* during *Aeromonas hydrophila* infection

Mehar Un Nissa,[1] Nevil Pinto,[2] Biplab Ghosh,[3] Anwesha Banerjee,[4] Urvi Singh,[5] Mukunda Goswami,[2] Sanjeeva Srivastava[1]

**ABSTRACT** In the aquaculture sector, one of the challenges includes disease outbreaks such as bacterial infections, particularly from *Aeromonas hydrophila* (*Ah*), impacting both wild and farmed fish. In this study, we conducted a proteomic analysis of the intestinal tissue in *Labeo rohita* following *Ah* infection to elucidate the protein alterations and its implications for immune response. Our findings indicate significant dysregulation in extracellular matrix (ECM)-associated proteins during *Ah* infection, with increased abundance of elastin and collagen alpha-3(VI). Pathway and enrichment analysis of differentially expressed proteins highlights the involvement of ECM-related pathways, including focal adhesions, integrin cell surface interactions, and actin cytoskeleton organization. Focal adhesions, crucial for connecting intracellular actin bundles to the ECM, play a pivotal role in immune response during infections. Increased abundance of integrin alpha 1, integrin beta 1, and tetraspanin suggests their involvement in the host's response to *Ah* infection. Proteins associated with actin cytoskeleton reorganization, such as myosin, tropomyosin, and phosphoglucomutase, exhibit increased abundance, influencing changes in cell behavior. Additionally, upregulated proteins like LTBP1 and fibrillin-2 contribute to TGF-β signaling and focal adhesion, indicating their potential role in immune regulation. The study also identifies elevated levels of laminin, galectin 3, and tenascin-C, which interact with integrins and other ECM components, potentially influencing immune cell migration and function. These proteins, along with decorin and lumican, may act as immunomodulators, coordinating pro- and anti-inflammatory responses. ECM fragments released during pathogen invasion could serve as "danger signals," initiating pathogen clearance and tissue repair through Toll-like receptor signaling.

**IMPORTANCE** The study underscores the critical role of the extracellular matrix (ECM) and its associated proteins in the immune response of aquatic organisms during bacterial infections like *Aeromonas hydrophila*. Understanding the intricate interplay between ECM alterations and immune response pathways provides crucial insights for developing effective disease control strategies in aquaculture. By identifying key proteins and pathways involved in host defense mechanisms, this research lays the groundwork for targeted interventions to mitigate the impact of bacterial infections on fish health and aquaculture production.

**KEYWORDS** *Aeromonas hydrophila*, extracellular matrix, focal adhesion, gut proteomics, mass spectrometry, rohu

Aquaculture stands out as one of the most rapidly expanding sectors, serving as a crucial source of high-quality protein to ensure global food security and foster international trade. However, the intensive aquaculture sector faces challenges such as disease emergence and stress factors, demanding careful consideration for sustainable production goals. Fish frequently encounter pathogens in environment comprising

Address correspondence to Sanjeeva Srivastava, sanjeeva@iitb.ac.in, or Mukunda Goswami, mukugoswami@gmail.com.

Nevil Pinto and Biplab Ghosh contributed equally to this article.

The authors declare no conflict of interest.

See the funding table on p. 15.

viruses, bacteria, fungi, and parasites. The primary disease causing bacteria in carps belong to *Aeromonas* spp., a highly opportunistic pathogen accounting for 66.66% of bacterial infections in many fish types including carps, salmon, trout (1).

Bacterial infections cause high mortality rates in both wild and farmed fish (2). The most commnonly encountered strains are *Aeromonas hydrophila*, *Aeromonas caviae*, *Aeromonas veronii*, *Aeromonas salmonicida*, and *Aeromonas sobria*. Among these bacteria, *A. hydrophila* has been considered as the most harmful pathogen to aquatic species, mainly causing hemorrhagic disease in fish farms (3). These do not get detected until diagnosis of symptoms associated with diseases like aeromonad septicemia/bacteremia and gastroenteritis are observed. The oral route serves as the predominant mode of entry, exposing the gut to potential infections. Successful manifestation of disease requires bacteria to adhere, invade, and colonize the host, evading immune responses through various mechanisms and substrates (4, 5). The tissue specificity of bacterial invasion, particularly in the gut, underscores the critical role of the extracellular matrix (ECM) as a facilitator of infection. The ECM, a fundamental cellular component, not only provides structural support but also plays a vital signaling role (6). It creates a microenvironment for essential signals that govern immune cell migration, activation, proliferation, and differentiation within infected tissues (7). Each marked tissue-specific pathogenic infection has a distinct ECM signature. Different tissue-specific infections exhibit distinct ECM signatures, comprising structural matrix components like collagen, proteoglycans, glycosaminoglycans, and functional elements such as enzymes and growth factors (8). The enzymes repair and remodel the structural components by enhanced synthesis and deposition, degradation, and protein modification, ultimately impacting the overall microenvironment essential for the infection and inflammatory response. The breakdown of primary and secondary defenses by bacterial enzymes, such as collagenases and hyaluronidases, significantly affects overall immunity (9).

A peek into the dynamics of the unidentified multifactorial pathogenesis of this bacterium is offered by omics studies ranging from genomics to transcriptomics and proteomics, enabling better understanding and development of effective disease control strategies. Jin et al. have constructed the complete genome sequence of virulent strain, *A. hydrophila* HX-3, and drawn its comparison to other strains in order to identify key genes responsible for virulence and quorum sensing (10). Another team, employing BOX-PCR-based DNA fingerprinting, evaluated banding patterns to investigate the strain-level genotypic markers in *Aeromonas hydrophila* (*Ah*) (11). The study by Hu et al. delve through transcriptomics and proteomics to identify mucosal immunity, i.e., first line of defense in gills of *Carassius auratus* against *Ah* (12). With the tangent of liver tissue analysis, a proteomic study suggests that *Ah* infection modulates host metabolic pathways and innate immunity with identified markers such as Toll-like receptors and C-lectins in *Labeo rohita* (13). Our study builds upon these efforts by employing holistic discovery-based and validation-based proteomic methods to unravel the pathogenesis of *A. hydrophila* in the intestinal tissue of *Labeo rohita (rohu)*. This species is a significant Indian carp, known for its nutritional value and widespread consumer preference, and ranks among the top 10 inland fish produced globally (14). We have provided an overall analysis of the relative changes observed in protein profiles pertaining to intestine tissue and have narrowed our findings to ECM-dominant inferences. This has enabled us to identify pathways and markers that if perturbed can be used for both disease diagnosis and treatment of clinical manifestations.

## MATERIALS AND METHODS

### Overall experimental design

This study involved a proteomic analysis of intestine tissue in *Labeo rohita* following infection with *Ah*. Fish with an average weight of 70 ± 10 g and a length of 19 ± 1 cm were infected with *Ah* and then sampled, resulting in a total of 12 samples, with six from the control group and six from the *Ah*-infected group (AH). From these

samples, four from each group were selected for discovery-based proteomic analysis. The MaxQuant software was used to analyze the data, followed by statistical analysis in the MetaboAnalyst tool to identify the differentially expressed proteins (DEPs; $P$ value 0.05, fold change 1.5). An overview of the functional annotation of altered proteins and metabolic pathways was obtained by gene ontology (GO) analysis. The SRM technique was employed to validate the protein abundance changes for a panel of differentially expressed proteins using all 12 samples.

## Maintenance of fish for the experiment

Six-month-old fish ($N$ = 150; average weight 70 ± 10 g) were obtained from a nearby fish farm in the Pen Raigad District of Maharashtra and transported to a wet lab facility (ICAR-CIFE, Mumbai, India) for the study. In three circular fiber tanks, the fish were evenly spaced, acclimated, and fed 2% of their body weight. The fish were maintained in a temperature range of 26°C to 28°C with adequate aeration and daily fecal removal. Every fish was checked for external clinical symptoms, and a random subset of fish from each of the tanks was slaughtered in order to look for pathogens in the fish using PCR.

## Bacterial challenge and tissue sampling

Bacterial isolation and identification were done as reported in our previous work (13). In brief, tissue from a naturally co-infected fish was used for isolation of $Ah$ strain (NCBI accession no. MT374248), which was confirmed using PCR. Fish were acclimatized and starved for 2 days followed by distribution into six high-density polyethylene plastic crates, each holding six fish, kept at 26°C–28°C. Three of the crates labeled as AH group were intraperitoneally inoculated with an LD$_{50}$ dose of 1.5 × 10$^8$ bacterial cells in phosphate-buffered saline (PBS). Three crates of the control group were subjected to equivalent volume of PBS injection. Following infection, there were visible signs of hemorrhage in the AH group, whereas the control group did not exhibit these symptoms, as expected. Intestine (midgut and hindgut) samples were collected 48 hours post challenge and washed in PBS two to three times to remove any remaining food particles that might be present in the samples. This ensures that the samples are clean and not contaminated with external materials that could interfere with subsequent analyses. Collected samples were stored at −80°C until further use. Tissue from three fish was combined into one yielding a total of six samples each for the control and AH groups. The samples were designated as Gut-C1 to C6 and Gut-AH1 to AH6, respectively.

## Tissue lysate preparation and protein extracts

Sodium dodecyl sulfate (SDS)-containing lysis buffer (5% SDS, 100 mM Tris-HCl [pH 8.5] [adjusted with phosphoric acid]) was used to prepare the tissue lysates. The tissue was weighed (40–50 mg) and washed in 1× PBS solution. Following it, 250 µL of lysis buffer and 5 µL of protease inhibitor cocktail (50× stock, Sigma—catalog no. 11873580001) were added to the tissue. It was then left to sit on ice for 30 minutes. Sonication was done for 2 minutes at a 40% amplitude and pulse 5 s on/off. The debris was removed using centrifugation, and the clear supernatant was collected.

## Quantification and digestion of proteins

Tissue protein quantification and digestion were done as mentioned in our previous work (13). In brief, protein was quantified through bicinchoninic acid (BCA) protein assay using bovine serum albumin as standard. Before digestion, 30 µg of protein (in SDS lysis buffer) was reduced with 20 mM TCEP. Reduced protein was loaded onto a 30-kDa column for downstream steps of alkylation and digestion. Iodoacetamide was used for alkylation. For digestion, trypsin in 50 mM ABC in a 1:30 ratio with protein was added over the column itself and kept in a wet chamber at 37°C for 16 hours. Peptides were dried and stored. Before mass spectrometry analysis, peptides were cleaned using in-house-prepared C18 tips (66883-U, Merck).

## Data acquisition with liquid chromatography tandem mass spectrometry

Peptides were quantified for each sample using the Scopes method (15). Following quantification, 1 µg of each peptide sample was injected to the mass spectrometer for liquid chromatography with tandem mass spectrometry (LC-MS/MS) analysis. Eight samples were run with a 120-minute LC gradient with a flow rate of 300 nL/minute. The mobile phase included solvent A as 0.1% formic acid (FA) and solvent B as 80% acetonitrile (ACN) with 0.1% FA. Samples included four each for the control and AH groups, respectively. An Easy-nLC nano-flow liquid chromatography 1200 system linked with an Orbitrap-Fusion Tribrid mass spectrometer was used to generate the data in a data-dependent acquisition mode. Peptides were loaded onto the trap column (Thermo Fisher Scientific, 100 mm × 2 cm, nanoViper C18, 5 mm, 100A) at a flow rate of 5 µL/minute and then passed through the analytical column (Thermo Fisher Scientific, 75 µm × 50 cm, 3 µm particle, and 100 Å pore size). MS1 parameters included Orbitrap resolution 60 k, scan range 375–1,700, RF Lens 60%, maximum injection time 50 s, exclusion duration 40 s, and mass tolerance 10 ppm. MS2 settings were isolation mode Quadrupole, isolation window 2, activation type HCD, Orbitrap resolution 15 k, maximum injection time 30 ms, and data type centroid. For MS1 and MS2 levels, the AGC target was set at 400,000 and 10,000, respectively. For positive internal calibration, a lock mass of 445.12003 $m/z$ was used.

## Protein identification and label-free quantification

The raw data obtained from LC-MS/MS were searched in MaxQuant (v1.6.6.0) software through Andromeda search engine. Under group-specific parameters, multiplicity was set as standard and labeled as 1. For modifications, oxidation at methionine (+15.994915 Da) and acetyl (protein N term) was chosen. Carbamidomethyl (C) was set under fixed modifications. Data were analyzed in label-free quantification mode, and match between runs was enabled. Trypsin/P was chosen as protease, and a maximum of two missed cleavages was allowed. Instrument type was selected as Orbitrap keeping all parameters as default. Under global parameters tab, the *Labeo rohita* UniProt whole protein sequence (UP000290572, ID—84645) database was selected and the include contaminant option was enabled. The minimum peptide length was set to 7. Maximum precursor mass tolerances were set at 20 ppm in the first search, 4.5 ppm in the main search, and 20 ppm for fragment mass tolerances. The false discovery rate for proteins, peptides, and PSMs was 1%. Proteins were only recognized by their unique peptide, and reverse was chosen as the decoy mode option. For a quantitative comparison, the label-free quantification (LFQ) intensities obtained for each sample were taken into consideration (Table S1).

## Statistical analysis using MetaboAnalyst

Statistical analysis was performed using MetaboAnalyst software, with MaxQuant output files as input (16). After MaxQuant analysis, the data set comprised the identified proteins, along with LFQ intensities for four samples each from control and AH groups. The data were filtered to remove features matched with contaminant or reverse sequences. The missing value imputation was done separately for control and AH groups using the KNN feature wise option, and features with more than 30% missing values were removed. Further statistical analysis was performed taking the common features between control and AH groups. No data filtration was applied, and data were log10 transformed before being plotted. The fold change threshold was taken to be 1.5, and RAW $P$ value cutoff was 0.05 to get the DEPs (Table S1). Computation of the variable importance in projection (VIP) score was done via the partial least squares discriminant analysis (PLS-DA), which is a supervised method. As a weighted sum of squares of the PLS weight, VIP indicates the importance of the variable to the whole model. Heatmap representing the expression of top DEPs was also obtained from MetaboAnalyst analysis. Volcano plots were plotted using online tool VolcanoseR (17). These significant DEPs

were further validated by referencing existing literature on their functions and roles in microbial diseases.

## Gene ontology, pathway, and protein-protein interaction enrichment analysis

Significant DEPs were considered for functional annotation and biological pathway analysis. Gene names of dysregulated proteins were retrieved from EggNOG (18) resource on the basis of ortholog annotation and from literature (as the gene names for *L. rohita* are not updated yet in the available databases). Metascape tool was used to obtain statistically enriched terms (GO/Kyoto Encyclopedia of Genes and Genomes pathway [KEGG] terms, canonical pathways) for all DEPs. STRING tool version 11.5 was used for protein-protein interaction (PPI) and visualization for the selected list of proteins from interested GO/KEGG terms/canonical pathways (Table S2). A minimum overlap of 3, a $P$ value cutoff of 0.05, and a minimum enrichment of 1.5 were employed.

## Validation of differential protein abundance using targeted proteomics

The proteomic validation was done using the selected/multiple reaction monitoring (SRM/MRM) method. The SRM method was created using transition list obtained from Skyline version 23.0.9.187. The Skyline setting included missed cleavage as 0 and precursor charges as +2 and +3, and product charge was set at +1 with "y" ion transitions (from ion 2 to last ion −1). Initial optimization involved 12 upregulated proteins with a pooled peptide sample tested against six lists each with 350–400 transitions. For data acquisition, TSQ Altis mass spectrometer (Thermo Fisher Scientific, USA) coupled to an high-performance liquid chromatography (HPLC)-Dionex Ultimate 3000 system (Thermo Fisher Scientific, USA) was utilized. Hypersil Gold C18 (1.9 µm, 100 × 2.1 mm, Thermo Fisher Scientific, USA) reverse-phase column was used for peptide separation. A binary buffer system with a flow rate at 0.45 mL/minute was employed for 10 minutes with buffers A and B constituting 0.1% FA and 80% ACN in 0.1% FA, respectively. For final runs, samples were spiked-in with an equal amount of heavy R labeled synthetic peptide ENQTCDIYNGEGR to ensure consistency across the runs. Including 10 transitions for synthetic peptide, the final method contained 402 transitions (Table S3). One microgram of peptides from 12 samples, 6 each for control and AH groups, was injected against the final method.

## Targeted proteomics data analysis

The result files (.raw) were imported to Skyline software and assigned to conditions of control and AH, respectively. The data-dependent acquisition (DDA) spectral library for the analysis of SRM data was built using the MSMS (.msms) file obtained from MaxQuant analysis. Data refinement was manually carried out considering the peak shape, retention time alignment, and dot product-based match with spectral library. The peptides not adhering to norms and specifications were deleted in order to refine the data. Statistical analysis was performed using the MSstats external tool inbuilt in Skyline to identify peptides with significant fold change (cutoff 1.5) and $P$ value (≤0.05). Result reports for all the peptides containing their peak area values were exported in .csv format to carry out the further analysis (Table S3).

## RESULTS

### Differential intestinal proteome between control and *Ah*-infected groups and overall functional annotation

Discovery-based proteomics data were acquired for eight samples to compare the proteomes between control and AH (48 hours post *Ah* infection) groups. Label-free quantification was used to compare the proteome profiles between infected and uninfected intestine tissues (details in Materials and Methods and the workflow shown

in Fig. 1). The LFQ analysis resulted in the identification of 2,632 proteins (features). After missing value imputation, the processed data included 1,961 and 1,744 as respective features for control and AH with 1,638 features accounting for the overlap (Table S1). Statistical analysis of the common features resulted in the identification of 148 DEPs of which 74 were upregulated and 74 were downregulated in the AH group as compared with the control group (Fig. 2A; Table S1). Using the PLS-DA model based on the VIP score, some features could distinguish the AH group from the control group. Figure 2B shows top 20 features classifying the AH and control groups based on VIP scores. Furthermore, the samples were clearly clustered with similar abundance profiles using unsupervised hierarchical clustering, which also stratified the *Ah*-infected and control groups as represented for top 25 features (Fig. 2C). A few of the DEPs including

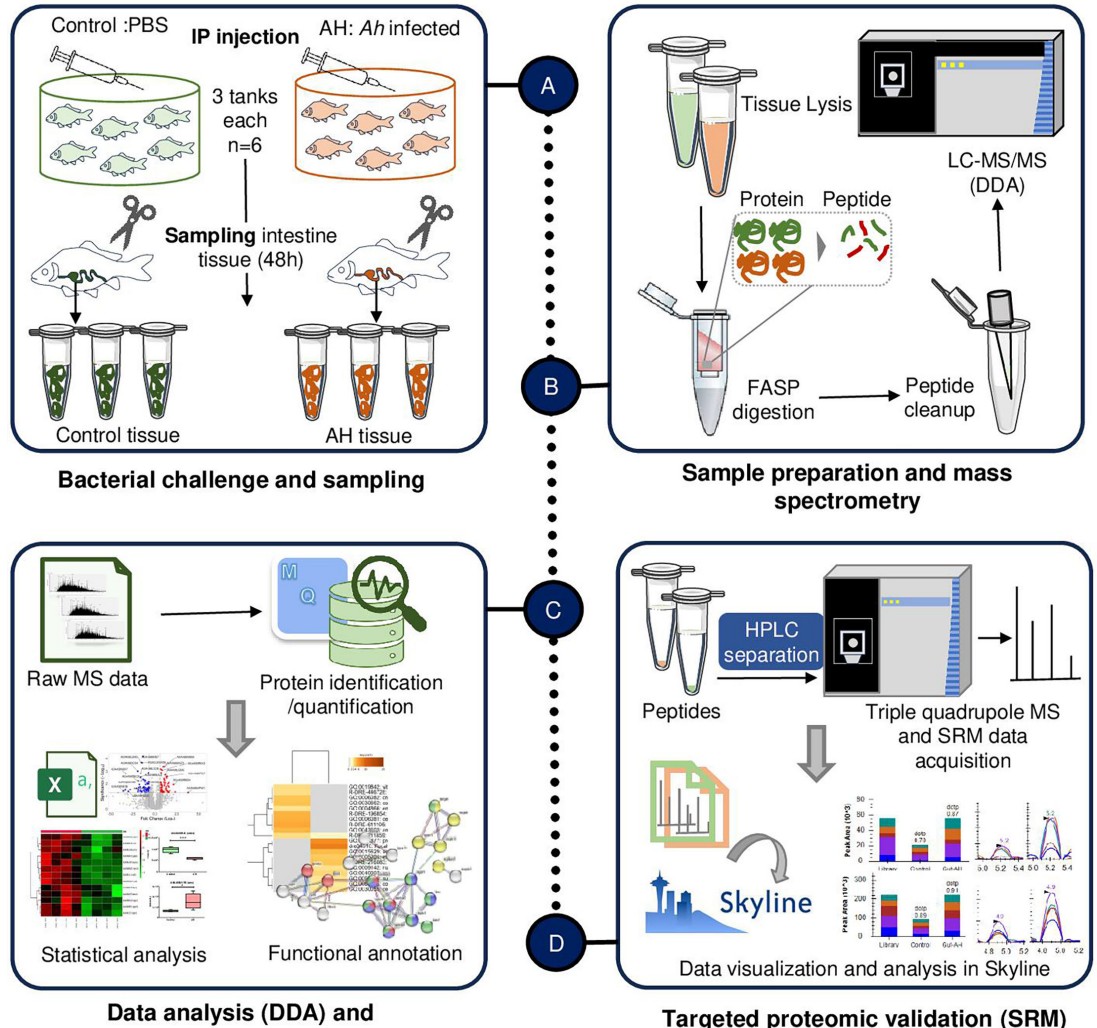

**FIG 1** Schematic representation of the study. (A) In the challenge study, the control group received intraperitoneal injections of PBS, while the AH group (*Ah* challenged) received *A. hydrophila* injections. After 48 hours of infection, fish were euthanized, and intestine samples were collected, as described in the text. (B) For proteomics analysis, tissues were lysed in a SDS-containing buffer, and protein digestion was carried out using the filter-assisted sample preparation (FASP) method. Subsequently, peptide samples underwent cleaning before being subjected to mass spectrometry for DDA using Orbitrap mass spectrometry. (C) The acquired raw mass spectrometry data (.raw) underwent analysis with MaxQuant software for protein identification and quantification. Statistical analysis was performed to identify differentially expressed proteins, followed by functional analysis. (D) Targeted proteomic validation of selected proteins through selected reaction monitoring involved subjecting peptide samples to HPLC. This was followed by target precursor and transition selection in the triple quadrupole mass spectrometer for the acquisition of spectral data. The SRM data were analyzed using Skyline software, and the targeted data were compared with the spectral library.

tetraspanin (tspan), elastin (eln), and annexin (Anx) among upregulated proteins are shown (Fig. 2D through F) and antithrombin-III (Serpinc1), hyaluronan-binding 2 (Habp2), and peroxiredoxin-like 2 (Prxl2a) for downregulated proteins (Fig. 2G through I) in the AH group compared with the control group.

Functional mapping of upregulated and downregulated DEPs was done using Metascape to understand how *Ah* infection influences on the host proteome

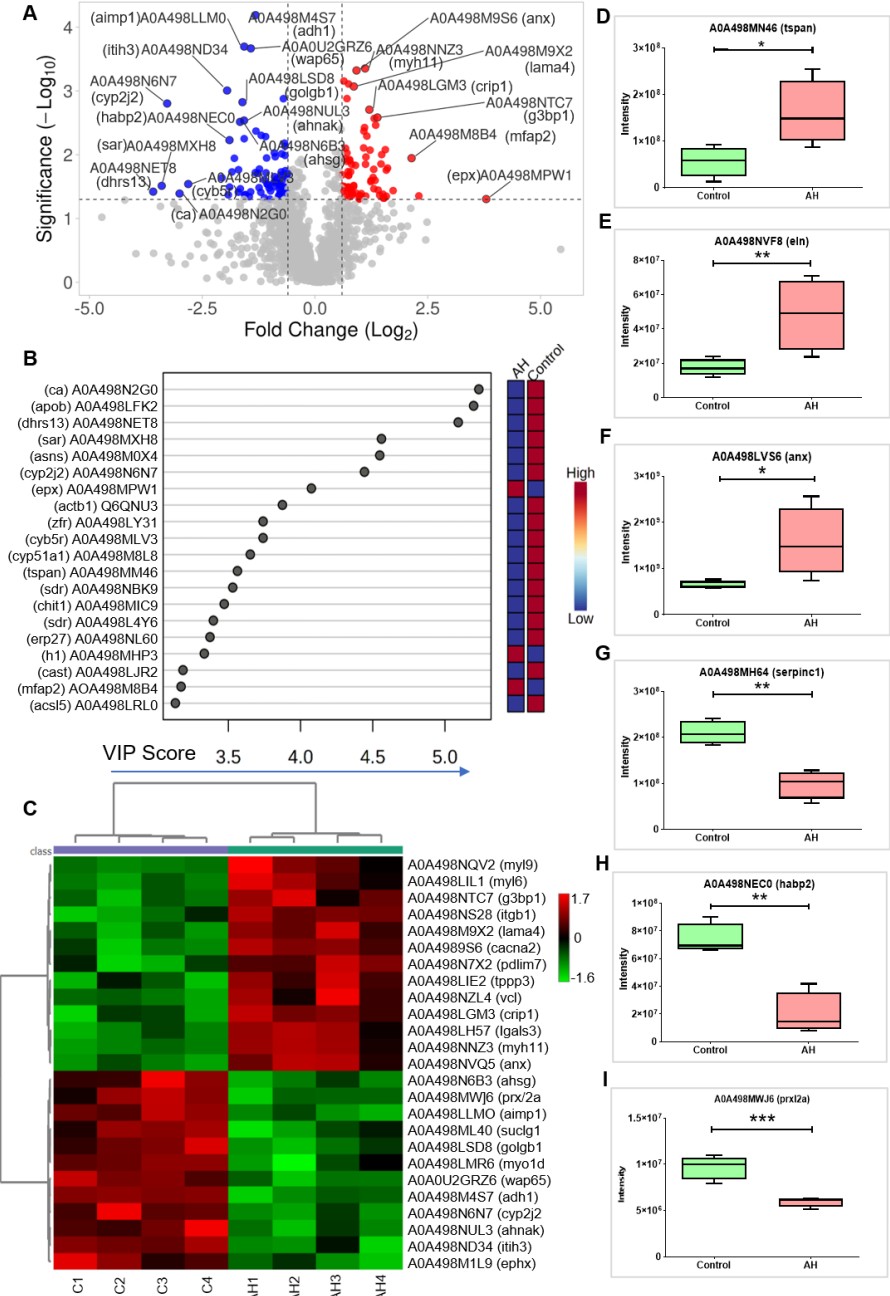

**FIG 2** Shotgun proteomic analysis shows altered host intestinal proteome during *Ah* infection. (A) Volcano plot depicting significantly altered protein candidates (fold change of 1.5, *P* < 0.05), with red and blue colors representing upregulated and downregulated proteins, respectively. (B) Top 20 altered key proteins found on the basis of VIP score. (C) Heatmaps showing differential expression (abundances) of top 25 significant proteins (*t*-test, *P* ≤ 0.05) across four replicates each of control tissues (C1 to C4) and AH-infected tissues (AH1 to AH4). (D–I) Box plots representing altered abundances of six proteins (three upregulated and three downregulated) in *Ah*-infected group as compared with control group (for box plots, *P* ≤ 0.05, **P* ≤ 0.01, and ***P* ≤ 0.001).

qualitatively. Figure 3A shows the significant GO terms and pathways mapped to the DEPs. For downregulated DEPs, the mapped GO terms include vitamin binding, organic acid metabolic process, endopeptidase inhibitor activity, cellular aldehyde, and amide metabolic process. The mapped pathways include cell junction organization, metabolism of vitamins and cofactors, and respiratory electron transport. Conversely, GO/pathways mapped to upregulated DEPs include protein containing complex binding, actin cytoskeleton, extracellular matrix structural constituent, supramolecular complex, focal adhesion, cell substrate junction, and integrin cell surface interactions (Fig. 3A; Table S2).

## Ah infection altered proteins related to focal adhesion and extracellular matrix pathways

As mentioned above, along with other pathways, enriched GO terms/pathways related to ECM were focal adhesion, actin cytoskeleton, integrin cell surface interactions, and extracellular matrix structural constituent. Enrichment of these pathways with high confidence might be an indication that ECM in the intestine plays an important role during *Ah* infection. In the focal adhesion pathway, 20 DEPs were mapped; similarly, 13 proteins were mapped to actin cytoskeleton, 5 to the extracellular matrix constituent, and 9 to the integrin cell surface interaction pathway (Fig. 3B through E). In total, 39 of unique significant upregulated DEPs were mapped to these processes/pathways highlighting the active focal adhesion and ECM signaling pathways in enterocytes. Furthermore, in the Metascape tool, the MCODE algorithm was applied on the enrichment network to identify neighborhoods where proteins are densely connected. Top MCODE network components for DEPs included protein-containing complex binding, focal adhesion, and supramolecular fiber organization (Fig. 3F through H).

PPI interaction analysis was performed for these ECM-related DEPs using STRING to obtain a network summarizing their functional associations. A total of 22 proteins formed protein networks (Fig. 4A). Integrins (itga1 and itgb1) formed the hub and had the highest number of connections with collagen (col6a3) and laminin proteins (lama5, lamb1, and lama4). Integrin (itga1) also showed interaction with myosin and tropomyosin proteins mapped to actin cytoskeleton. Collagen (col6a3) showed interaction evidence with decorin (dcn) and lumican (lum) (Fig. 4A).

## Targeted proteomic validation using the selected reaction monitoring approach

The refinement of SRM validation data in Skyline ended up with 58 peptides corresponding to 10 DEPs. On group comparison of the six replicates each from AH and control samples, the overall trend for protein expression was consistent with the DDA data in terms of abundance profile in *Ah*-infected (AH condition) samples. Furthermore, applying a cutoff criterion of a *P* value of 0.05 and a fold change of 1.5, 25 peptides corresponding to 8 DEPs showed significantly differential peak areas and intensities. Five of these DEPs belong to ECM-associated pathways. Three proteins showed significant results with ≥2 unique peptides. It included Itga1 (eight peptides), decorin (six peptides), and lumican (two peptides) showing significantly higher intensity in the AH group compared with the control group (Fig. 4B through D). Also, two proteins from the focal adhesion pathway, namely, Mcam (MUC-18 like isoform) and Mfap2 (microfibrillar associated 2 protein), had one significant upregulated peptide each in the validation data.

The validation experiment included three more proteins, which were upregulated in the discovery data. It included two annexin proteins (A0A498LVS6 and A0A498MM94) and GGT5 (gamma glutamyltransferase-5 like protein) (Table S3). Annexins showed upregulation in the AH group in the SRM data each with three unique peptides, and Ggt5 had one unique significant peptide, and this trend is at par with the discovery data.

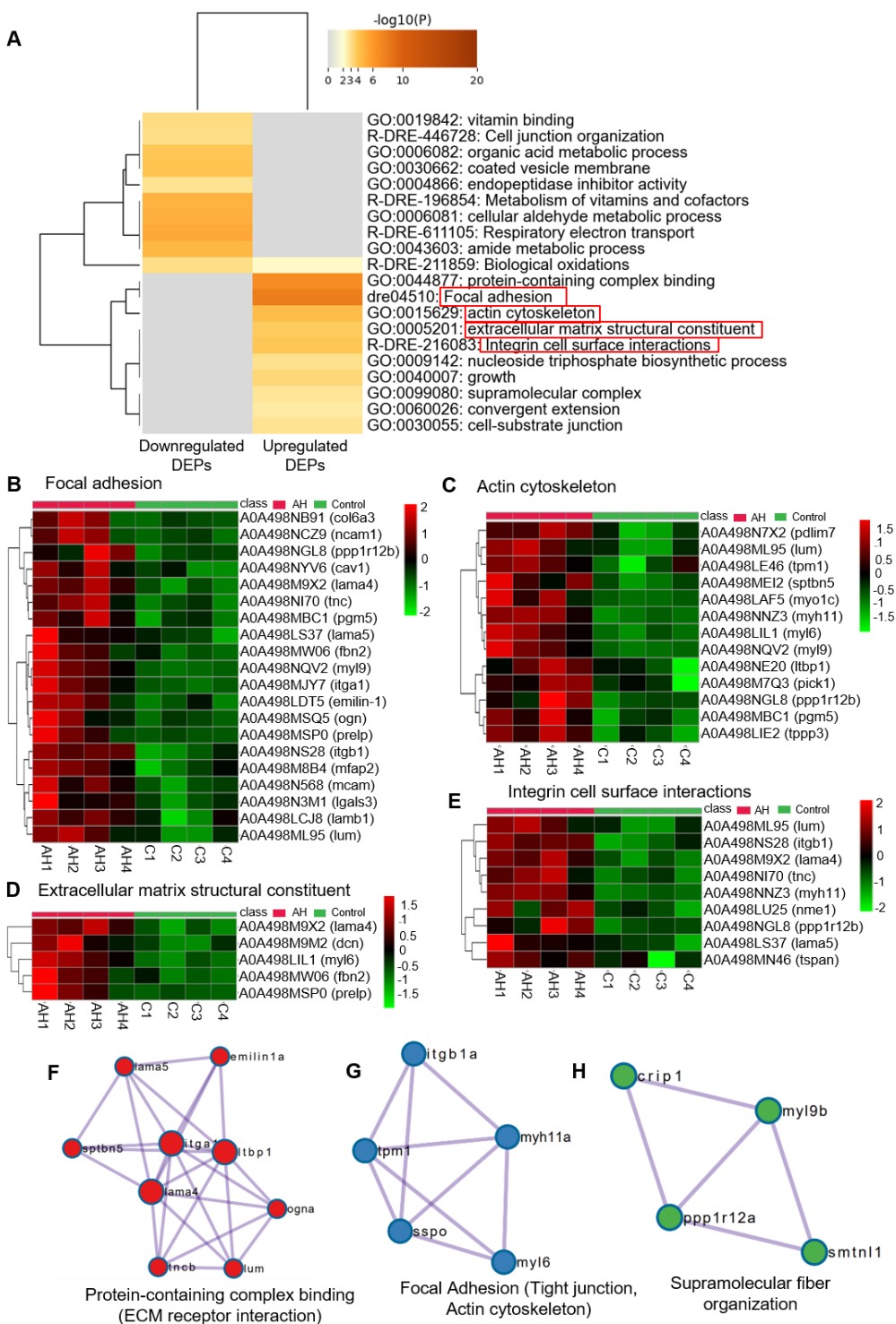

**FIG 3** GO and pathway analysis for differentially expressed proteins. (A) Hierarchical cluster of significant GO/KEGG terms based on kappa statistical similarities among their gene memberships for upregulated and downregulated DEPs. Then, 0.3 kappa score was applied as the threshold to cast the tree into term clusters. The term with the best *P* value within each cluster as its representative term is displayed in a dendrogram. The heatmap cells are colored by their *P* values, white cells indicate the lack of enrichment for that term in the corresponding gene list. Important pathways related to ECM are highlighted with red rectangle. (B–E) Heatmaps showing differential abundances of proteins mapped to focal adhesion pathway, actin cytoskeleton, extracellular matrix structural constituent, and integrin cell surface interaction pathway, respectively. (F–H) Top three Molecular Complex Detection (MCODE) network components for DEPs viz. protein-containing complex binding, focal adhesion, and supramolecular fiber organization.

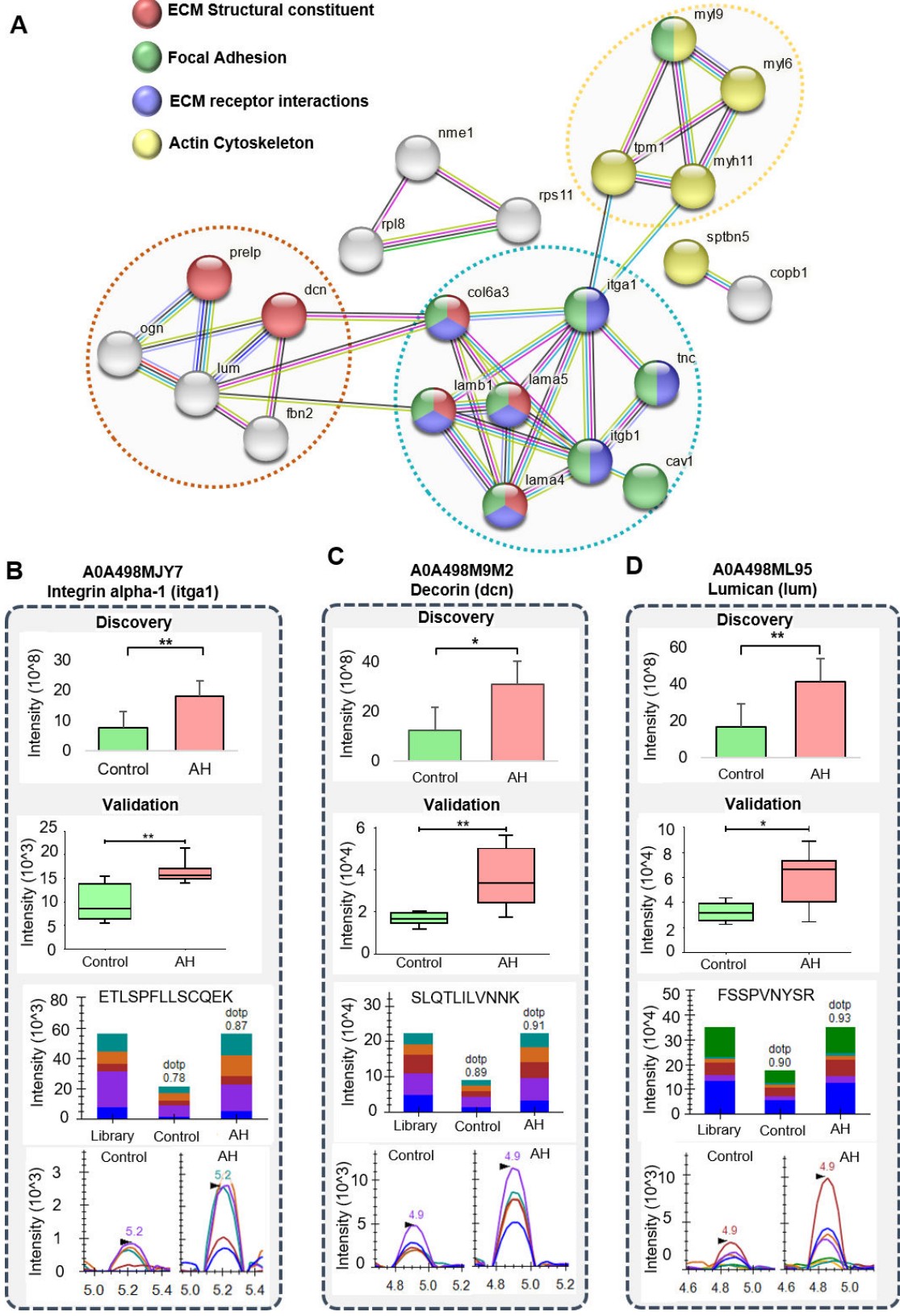

**FIG 4** Protein-protein interaction network for ECM-related pathways and validation of few candidates using targeted proteomics. (A) PPI network for proteins belonging to ECM structural constituent, focal adhesion, ECM receptor interactions, and actin cytoskeleton that showed interactions and three major networks were observed (encircled with dotted line). (B–D) Showing differential protein abundance in discovery and targeted experiment for three ECM-associated

Fig 4 (Continued)

proteins itga1, decorin, and lumican, respectively, upregulated in the AH condition (*Ah*-infected group) compared with the control condition. (B–D) The panel (up to down) represents the protein-wise intensities based on the shotgun analysis (DDA abundance) followed by SRM analysis. *$P ≤ 0.05$ and **$P ≤ 0.01$. The latter half contains bar plots for peak area and peak view for intensity comparison, for a representative peptide of each protein with AH vs control condition [with dot product (dotp) value based on match with spectral library]. Different colors in the bar plots and spectral peaks represent different product ions of the same peptide.

## DISCUSSION

The extracellular matrix is well known for its role in immune signaling along with providing structural support to the cell. The diverse effects of the ECM can alter the invasion and dissemination of microbial pathogens in host tissues and greatly influence the overall immune response to infection (19). During tissue damage or infections, it responds by activating complex biochemical and physical systems that control intercellular communication, which is essential for preserving tissue homeostasis. A scaffold of biochemical and biomechanical signaling is created by the action of several actin cytoskeletal regulators, which are physically coupled to the extracellular matrix through integrins (20).

In the current study using the proteomic approach, we analyzed the overall changes in the intestinal proteome as a result of *Ah* infection. Several pathways and biological processes including respiratory electron transport, metabolism of vitamins, and amide and cell junction organization were affected. We found the dysregulation of ECM protein components in the intestine tissue of host (*Labeo rohita*) during *Ah* infection. Mechanical properties of microenvironments, such as ECM stiffness, can modulate the interaction of host cells and pathogen. ECM stiffness-regulated mechanical properties can promote apoptosis and alter the abundance of actin filaments (21) or increase the susceptibility of host cells to infection by bacterial pathogens (22). Highly organized matrix fibers and collagen/elastin cross-linking are among the critical determinants of ECM stiffness (23). In this study, we found increased abundance of elastin (2.7-folds) and collagen alpha-3(VI) chain (1.7-fold) in the AH group. Elevation in the level of the collagen alpha-3(VI) chain has been reported in serum of patients with gastrointestinal disorders (24). It was also found upregulated in intestine tissue of tongue sole *Cynoglossus semilaevis* after *Vibrio vulnificus* infection (25). This might indicate that *Ah* infection mediates changes in ECM composition that causes matrix rigidity.

To further explore the effect of protein dynamics during *Ah* infection, we looked into the statistically enriched terms/pathways (GO/KEGG terms) mapped to the significant DEPs, specifically those associated with ECM, actin bundles, and focal adhesions. Interestingly, the enriched pathways included focal adhesions (dre04510) and integrin cell surface interactions (R-DRE-216083) while actin cytoskeleton (GO:0015629) and extracellular matrix structural constituent (GO:0005201) were among the significant GO terms. Focal adhesions are macromolecular sites, which form mechanical connections between intracellular actin bundles and ECM through clustered integrin receptors. Focal adhesions are associated with the immune system and play an important role during infections or diseases. It has been reported that pathogens exploit focal adhesions to ensure their uptake, survival, and dissemination (26). They serve as scaffolds for many signaling pathways triggered by cell surface adhesion molecules, receptors as integrin, or a mechanical tension on the cell.

Integrins are alphabeta heterodimeric integral proteins and members of the cell adhesion receptor superfamily expressed in all metazoans. They are referred as the key molecules in cell-microenvironment and cell-cell communication. It was reported that expression of integrin beta was increased in *Penaeus monodon* after challenging with bacteria *Vibrio harveyi* and *Vibrio anguillarum*. Furthermore, the authors reported that inhibition of integrin beta gene caused downregulation of innate immunity-related genes (27). We also found increased abundance of integrin beta 1 (3.3-fold) and integrin alpha 1 (2.3-fold) in the AH group, in the discovery and the validation experiment.

Another important protein from the integrin cell surface interaction pathway was tspan, which showed a 2.9-fold increase during infection in our study. Tetraspanins represent a conserved superfamily of four-span membrane proteins, which are associated with innate immune response and bacterial adhesion as well (28). In a study on giant freshwater prawn, *Macrobrachium rosenbergii* infected with *Ah*, the expression of tspan8 (tetraspanin 8) was increased in hepatopancreas and gill tissue. According to the authors, pre-incubation of peptides from the long extracellular loop of tspan8 protein decreased the apoptosis caused by *Ah* infection in prawn tissue (29). These results suggest that integrins and tspan may play a role in the host response to *Ah* infection, consistent with findings from similar prior studies. However, further research is needed to confirm their specific functions.

Subunits of myosin (myl6, myl9, and myo1c), tropomyosin (tpm1), phosphoglucomutase (pgm5), a protein from tubulin polymerization-promoting protein family (tppp3), lumican (lum), and a few more proteins showed increased abundance in the AH group and mapped to actin cytoskeleton (GO:0015629). These proteins control the cytoskeletal reorganization during infection and lead to changes in cell behavior. Latent-transforming growth factor beta-binding 1-like isoform X1 (ltbp1), which is 1.8-fold increased during *Ah* infection, was also mapped to actin cytoskeleton. It has been reported that LTBP-1 forms a complex facilitated by disulfide linkage with TGF-β propeptide (known as LAP or latency-associated peptide) in the endoplasmic reticulum before secretion. This binding assists in proper folding and release of large latent complex and its interaction with ECM. Transforming growth factor (TGF-β) is a crucial immunoregulatory cytokine that regulates cell proliferation, differentiation, survival, migration, and apoptosis in both healthy and pathological circumstances. TGF-β1 has been found upregulated with poly I:C or lipopolysaccharide (LPS) induction in *Culter alburnus*, a teleost (30). LTBP1 is reported to interact with fibrillin (and/or fibronectin) and other matrix components, which helps in the release of TGF-β from the latent complex to bind to its receptor (31). We identified fibrillin-2-like isoform X1 (fbn2) in this study, a structural constituent of extracellular matrix, mapped to focal adhesion and showed a 2.9-fold increased abundance during infection.

It has been reported that the first and the most important step towards the successful establishment of infection is the adherence of bacteria to the host cells. One crucial tactic is their interaction with host ECM proteins like fibronectin, collagen, elastin, vitronectin, and laminin (32). We found increased abundance of laminin proteins, namely, laminin subunit alpha-4 (1.8-fold), laminin subunit alpha-5 (1.8-folds), laminin subunit beta-1 (1.7-folds), elastin (2.7-folds), and collagen alpha-3(VI) (1.7-folds). Laminins interact with integrins, collagens, and other ECM components that allow the assembly and integrity of basement membrane. Laminin interaction with other ECMs also influences the migration and the function of immune cells (33). Another protein, galectin 3 (lgals3), mapped to the focal adhesion pathway showed increased abundance in the AH group. Galectin 3 is a glycan-binding lectin, reported to function in host defense as an opsonin. Galectin 3 when bound to LPS promoted the bacterial phagocytosis by microglia, i.e., brain macrophages (34). The dysregulation of these ECM proteins in the study suggests that these candidates might be important players for pathogen invasion during *Ah* infection (Fig. 5A).

In an earlier study, the transcriptional level of tenascin-c was found increased in ulcerated areas during inflammatory bowel disease (IBD) in the murine model of IBD (35). Tenascin-C is a proinflammatory ECM protein and is upregulated with tissue injury and cellular stress. Tenascin-C interacts with receptors like TLR4 and integrins to initiate immunomodulatory effects. The subsequent interaction with immune cells results in the induction of soluble proinflammatory mediators, such as interleukins (IL-6, IL-8, IL-1β, and IL-18) and tumor necrosis factor (TNF) (35). Based on these findings and our results, it could be anticipated that the increased abundance of tenascin-C during *Ah* infection may be associated with an increase in proinflammatory cytokines and chemokines, potentially coordinating the activation and recruitment of innate immune cells. (Fig. 5B).

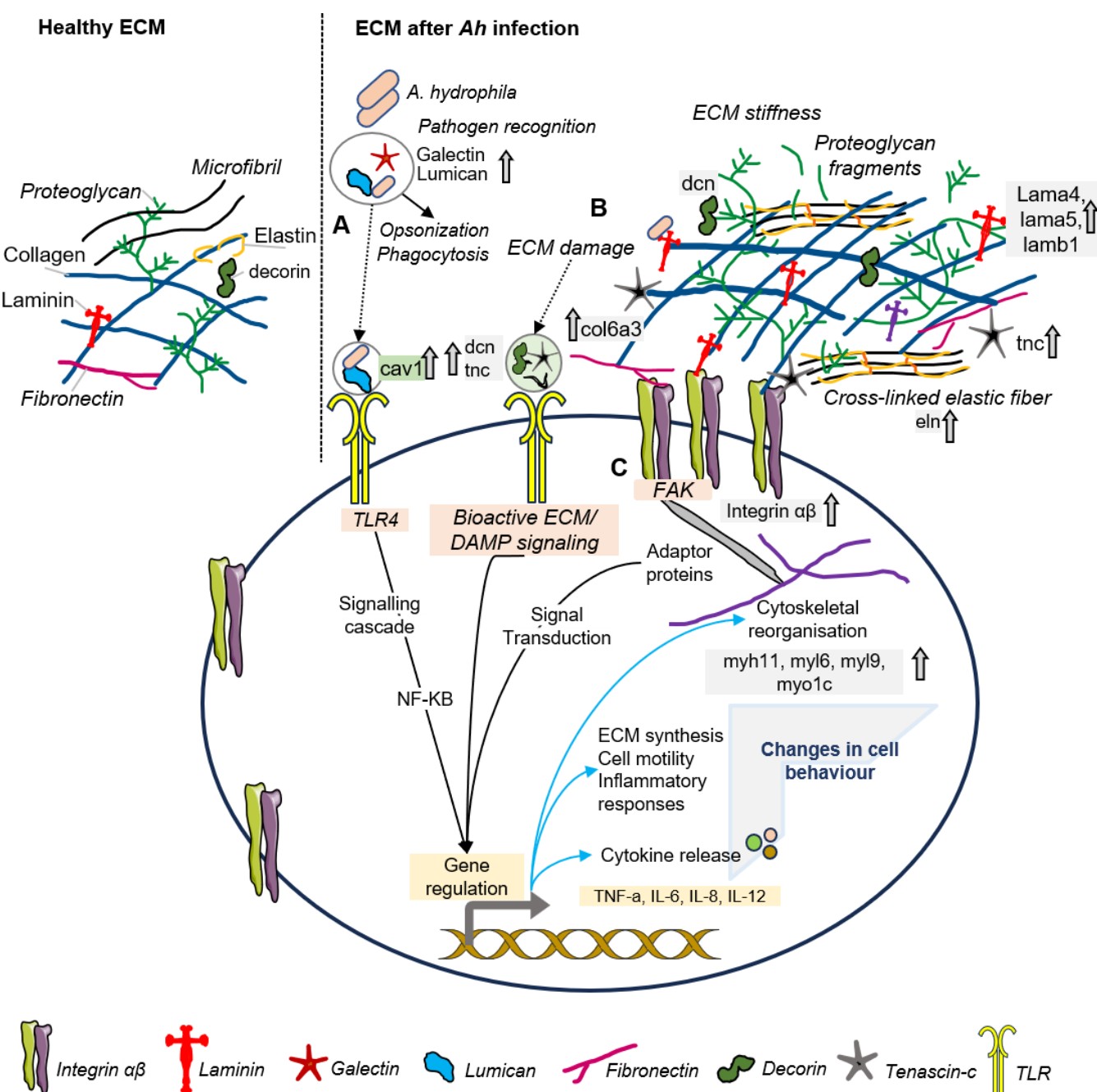

**FIG 5** Hypothesized interplay of *bacteria* and host tissue during *Ah* infection: left side shows the healthy ECM with normal architecture of ECM components and right side shows the compact and stiff ECM during *Ah* infection with cross-linked fibers and ECM fragments. (A) Interaction of the bacteria with the ECM proteins like fibronectin, galectin, lumican, and laminin helps in pathogen recognition and adherence with the host cells. These ECM proteins also act as opsonins to promote opsonization and phagocytosis. These proteins are also linked with TLR receptor-mediated cell response to affect gene regulation. (B) Once the pathogen is entered, it degrades the ECM through matrix-degrading enzymes resulting into bioactive ECM fragments. Bioactive ECM fragments and damage-associated molecular pattern (DAMP) immunomodulators as decorin (dcn), tenascin-c (tnc), and lumican further start cascade of signals by interacting with fibroblasts or surface receptors like TLRs on immune cells. As a result, more ECM proteins are produced and cross-linked ECM fibers are formed leading to stiffness. (C) Increased rigidity and stiffness in ECM followed by larger and stable focal adhesions lead to signaling through focal adhesion kinase and integrins. Such processes will further initiate mechanotransduction ending into inflammatory responses, release of cytokines, cytoskeletal reorganization, and changes in cell behavior for pathogen clearance and tissue repair. Upside arrow represents upregulated proteins in this study.

Once the pathogen has invaded and caused degradation in ECM, ECM fragments may be released to serve as "danger signals," i.e., DAMPs during infection, inflammation,

injury, or even aging. Such immunoregulatory fragments can be produced when the ECM is broken down by matrix-degrading enzymes (36). They lead a cascade of signaling, which results in the initiation of pathogen clearance and tissue repair. In this study, the AH group was found with higher abundance of few proteins, reported to act directly or indirectly as immunomodulators. Such proteins include decorin (2.5-fold), lumican (2.5-fold), tenascin-like isoform X1 (2.1-fold), and fibrillin-2 (2.9-fold). These ECM proteins are linked with the actin cytoskeleton and focal adhesion. Decorin is a small leucine-rich repeat proteoglycan, which coordinates pro- and anti-inflammatory cytokines and macrophage recruitment by directly binding to TLR2 and TLR4 receptors. Lumican is similar to decorin and known to bind with LPS and coordinates the innate immune response by enhancing LPS/TLR4-mediated proinflammatory responses (37). It has been reported that the circulating level of lumican was increased in human and mouse during sepsis and mice lacking lumican showed poor bacterial clearance. The author also reported that a protein caveolin (cav1) is required to maintain the lumican on cell surfaces (38). In this study, caveolin was increased by 1.6-fold in the AH group and mapped to the focal adhesion pathway.

With these immunomodulators, the signaling event may begin by interacting with pattern recognition receptors such as Toll-like receptors present on immune cells like dendritic cells and macrophages, as well as other cells like epithelial cells and fibroblasts (7, 26). This interaction could lead to the production of more ECM proteins and the formation of cross-linked ECM fibers (Fig. 5B and C), resulting in increased rigidity and stiffness in the ECM. These changes could subsequently lead to larger and more stable focal adhesions, thereby influencing focal adhesion signaling and altering cell behavior. Overall, our results suggest that these ECM proteins may play a role during *Ah* infection by potentially coordinating the initiation of the immune response through damage signals and their interaction with surface receptors such as TLRs and focal adhesion signaling pathways. However, further research is needed to confirm these mechanisms and their implications.

## Conclusions

The investigation into the proteomic profile of the intestine tissue in *Labeo rohita* following infection with *Ah* shows substantial alterations in the ECM proteome. Notably, 39 proteins associated with integrin cell surface interactions, focal adhesion, actin cytoskeleton, and extracellular matrix structural organization exhibit differential expression during *Ah* infection. This observation underscores the intricate interplay between the host's immune system and the ECM. The findings suggest a potential reciprocal relationship, wherein signals from the ECM play a crucial role in coordinating immune responses. Additionally, immune cells might contribute to ECM repair and regeneration through the release of cytokines, influencing ECM synthesis, cytoskeleton reorganization, and alterations in cell behavior. To our best knowledge, this is the first study to present landscape of intestinal proteome during *Ah* infection in this food fish. Our findings open avenues for future research in the field of aquaculture and disease control in *Labeo rohita* and related carps. Further investigations could focus on the specific mechanisms through which dysregulation of ECM proteins influences immune responses and host-pathogen interactions. Understanding the molecular signaling pathways during *Ah* infection could provide targets for therapeutic interventions.

### ACKNOWLEDGMENTS

This work was supported by the Department of Biotechnology (BT/PR15285/AAQ/3/753/2015) Govt. of India to S.S. and M.G.; M.U.N. was supported by University Grants Commission (UGC). We acknowledge the ICAR-Central Institute of Fisheries Education, Mumbai, for supporting this work. We acknowledge

MASS-FIITB at IIT Bombay supported by the Department of Biotechnology (BT/PR13114/INF/22/206/2015) for mass-spectrometric data acquisition.

M.U.N. did the following: conceptualization, methodology, investigation, validation, formal analysis, writing—original draft, review, and editing, and visualization: all original data. N.P. did the following: resource acquisition, formal analysis, writing—original draft, review, and editing and visualization. B.G. did the following: formal analysis, visualization, and writing—original draft, review, and editing. A.B. did the following: formal analysis, visualization, and writing—original draft, review, and editing. U.S. did the following: formal analysis, visualization, and writing—original draft, review, and editing. M.G. did the following: resources, conceptualization, and writing—original draft, review, and editing. S.S. did the following: supervision, resources, conceptualization, and writing—original draft, review, and editing.

## AUTHOR AFFILIATIONS

[1]Department of Biosciences and Bioengineering, Indian Institute of Technology Bombay, Mumbai, Maharashtra, India
[2]Central Institute of Fisheries Education, Indian Council of Agricultural Research, Versova, Mumbai, Maharashtra, India
[3]German Cancer Research Center (DKFZ), Heidelberg, Germany
[4]Indian Institute of Science Bangalore, Bangalore, Karnataka, India
[5]Friedrich Alexander University Erlangen Nuremberg, Erlangen, Germany

## PRESENT ADDRESS

Mehar Un Nissa, Institute for Systems Biology, Seattle, Washington, USA

## AUTHOR ORCIDs

Mehar Un Nissa ⓘ http://orcid.org/0000-0002-0581-1510
Mukunda Goswami ⓘ http://orcid.org/0000-0001-6863-7647
Sanjeeva Srivastava ⓘ http://orcid.org/0000-0001-5159-6834

## FUNDING

| Funder | Grant(s) | Author(s) |
| --- | --- | --- |
| Department of Biotechnology, Ministry of Science and Technology, India (DBT) | BT/PR15285/AAQ/3/753/2015,BT/PR13114/INF/22/206/2015 | Sanjeeva Srivastava |
| Department of Biotechnology, Ministry of Science and Technology, India (DBT) | BT/PR15285/AAQ/3/753/2015 | Mukunda Goswami |

## AUTHOR CONTRIBUTIONS

Mehar Un Nissa, Conceptualization, Formal analysis, Investigation, Methodology, Validation, Visualization, Writing – original draft, Writing – review and editing | Nevil Pinto, Formal analysis, Resources, Visualization, Writing – original draft, Writing – review and editing | Biplab Ghosh, Formal analysis, Visualization, Writing – original draft, Writing – review and editing | Anwesha Banerjee, Formal analysis, Visualization, Writing – original draft, Writing – review and editing | Urvi Singh, Formal analysis, Visualization, Writing – original draft, Writing – review and editing | Mukunda Goswami, Conceptualization, Resources, Writing – original draft, Writing – review and editing | Sanjeeva Srivastava, Conceptualization, Funding acquisition, Resources, Supervision, Writing – original draft, Writing – review and editing

## DATA AVAILABILITY

The protein database (.FASTA) and raw mass spectrometry data (.raw) have been deposited to the ProteomeXchange Consortium via the PRIDE partner repository. All result output files for protein identification are also submitted in text (.text) format along with the parameter file. Also, the spectral library (.blib) generated using the discovery data for analyzing the targeted data is uploaded. The identifier PXD037850 can be used to retrieve all of the data. The transition lists, skyline documents, and all SRM raw (.raw) data for the SRM experiment have been submitted to Panorama public and can be accessed through the link https://panoramaweb.org/rohugutproteomicsah.url.

## ETHICS APPROVAL

For this work, all fish were collected and sacrificed at The Indian Council of Agricultural Research, Central Institute of Fisheries Education (ICAR-CIFE), Versova, Mumbai, 400076. The presented work is part of the sanctioned project of the Department of Biotechnology, India (BT/PR15285/AAQ/3/753/2015), and the work was approved by the Institute Ethical committee, ICAR-CIFE (Project code 1008979).

## ADDITIONAL FILES

The following material is available online.

### Supplemental Material

**Table S1 (mSystems00247-24-s0001.xlsx).** Details of discovery data (DDA) used for LFQ analysis.
**Table S2 (mSystems00247-24-s0002.xlsx).** Details of pathways, gene ontology, and protein-protein interaction data analysis using Metascape and STRING.
**Table S3 (mSystems00247-24-s0003.xlsx).** Details of SRM data used for targeted proteomic analysis.

### Open Peer Review

**PEER REVIEW HISTORY (review-history.pdf).** An accounting of the reviewer comments and feedback.

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
