## [Reviewer comments · mSystems]

Proteomic Insights into Extracellular Matrix Dynamics in the Intestine of *Labeo rohita* During *Aeromonas hydrophila* Infection

Mehar Un Nissa, Nevil Pinto, Biplab Ghosh, Anwasha Banerjee, Urvi Singh, Mukunda Goswami, and Sanjeeva Srivastava

Corresponding Author(s): Sanjeeva Srivastava, Indian Institute of Technology Bombay

Review Timeline:

Submission Date:	February 18, 2024
Editorial Decision:	May 11, 2024
Revision Received:	July 9, 2024
Editorial Decision:	July 29, 2024
Revision Received:	August 3, 2024
Accepted:	August 9, 2024

Editor: Joshua Elias

Reviewer(s): Disclosure of reviewer identity is with reference to reviewer comments included in decision letter(s). The following individuals involved in review of your submission have agreed to reveal their identity: Leonard J. Foster (Reviewer #1)

Transaction Report:

DOI: <https://doi.org/10.1128/msystems.00247-24>

Re: mSystems00247-24 (Proteomic analysis of gut in *Labeo rohita* reveals ECM as Key Player in host's Response to *Aeromonas hydrophila* Infection)

Dear Prof. Sanjeeva Srivastava:

In addition to the comments from Reviewer 1, I would add the following:

- 1) The description of the tissue used for this study is too vague: was the entire GI (mouth through anus) from each fish used, or one region? How was GI contents removed?
- 2) The title, "Proteomic analysis of gut in *Labeo rohita* reveals ECM as Key Player in host's Response to *Aeromonas hydrophila* Infection" isn't precisely supported by the data from this paper. While the data presented here supports ECM as being involved in Ah infection response, they do not in and of themselves "reveal" this role. Simply relating enriched proteins from this small-scoped study to prior, more rigorous tests made by others who investigated other pathogens and other fish does not necessarily "reveal" that the same phenomena are relevant to the infection system investigated here. Some degree of orthogonal testing, such as the recommendation by Reviewer 1, would better support this notion. Relatedly, strong statements such as, "Such results indicate that integrins and tspan have important role in host response to Ah infection" (page 14) overstate the conclusions that can be drawn from the data the authors generated themselves. Again, linking them to similar prior studies can help them pose new hypotheses, but these links in and of themselves cannot be deemed proof of these hypotheses. Please rephrase this and similar statements throughout the document.

Revision Guidelines

Sincerely,
Joshua Elias
Editor
mSystems

Reviewer #1 (Comments for the Author):

This manuscript describes a proteomic assessment of the gut from a species of carp after intraperitoneal challenge with *Aeromonas*. The manuscript is clearly written and the functional validation has gone about as far as it is possible to go in this system, with one possible suggestion below. I particularly appreciate that the authors chose to validate their proteomics screen with SRM/PRM assays rather than Western blots.

I appreciate that there are few tools available in a non-model system such as this to do extensive follow-up. The only added experiment I could suggest is some kind of imaging or rigidity test to confirm that the extracellular matrix is, indeed, more rigid.

I wonder why the authors did not include *Aeromonas* sequences in the database search that they conducted? Did they reasonably believe that there would be no bacteria left? And what about the contribution of the gut contents to the proteins extracted? As far as I can tell in the methods, the gut contents would have been included in the preparation? It would be useful to know what other proteins might be present in these preparations, beyond the standard contaminants included in a MQ search.

Response to Reviewers

Manuscript: mSystems00247-24 (Proteomic analysis of gut in *Labeo rohita* reveals ECM as Key Player in host's Response to *Aeromonas hydrophila* Infection)

All comments are given due consideration and addressed to the best extent possible. Responses are *italicized in blue*. Text modified in the main document is written in red.

=====

Comments from the reviewer(s):

Reviewer #1 (Comments for the Author):

1). This manuscript describes a proteomic assessment of the gut from a species of carp after intraperitoneal challenge with *Aeromonas*. The manuscript is clearly written and the functional validation has gone about as far as it is possible to go in this system, with one possible suggestion below. I particularly appreciate that the authors chose to validate their proteomics screen with SRM/PRM assays rather than Western blots. I appreciate that there are few tools available in a non-model system such as this to do extensive follow-up. The only added experiment I could suggest is some kind of imaging or rigidity test to confirm that the extracellular matrix is, indeed, more rigid.

Response: *Thank you for the positive remarks and thorough review of our manuscript. We appreciate the suggestion to add a rigidity test to our analysis. The most common techniques for assessing matrix stiffness in biological research include Atomic Force Microscopy (AFM), Traction Force Microscopy (TFM), and Magnetic Resonance Elastography (MRE). Unfortunately, due to our lab's constraints and the unavailability of these facilities, we cannot perform these tests at this time. Additionally, outsourcing these tests is currently impractical due to limited funding.*

*Other techniques, such as confocal microscopy with fluorescent reporters, could indirectly assess matrix rigidity. This method involves using fluorescently labeled antibodies to target specific ECM components or cell-surface proteins, allowing us to visualize and infer mechanical properties. However, we were unable to perform these validations because specific antibody resources for *Labeo rohita* are not available. Molecular research on this species is still in its developmental stage, and the literature is limited. Developing new antibodies would be costly, time-consuming, and prone to failure.*

Given these limitations, we chose to validate our proteomics screen using Multiple/Selected Reaction Monitoring (M/SRM), a mass spectrometry-based approach. This technique selectively monitors peptides of target proteins and provides a reliable measure of protein expression. It is well-established, reliable, and feasible with our current resources, making it the best choice for our study without the use of antibodies.

We acknowledge the reviewer's suggestion and agree that incorporating rigidity tests would enhance our analysis. We plan to pursue these tests in future studies as part of the ongoing

Labeo rohita project. We will aim to include such analyses in follow-up studies and publish the results separately. Once again, we appreciate the valuable feedback and suggestions provided, which will help guide our future research efforts.

2). I wonder why the authors did not include *Aeromonas* sequences in the database search that they conducted? Did they reasonably believe that there would be no bacteria left? And what about the contribution of the gut contents to the proteins extracted? As far as I can tell in the methods, the gut contents would have been included in the preparation? It would be useful to know what other proteins might be present in these preparations, beyond the standard contaminants included in a MQ search.

Response:

*Thanks to the reviewer for this comment. We would like to state that the *Aeromonas* sequences were not included in this study as in this study we specifically aimed at host proteome dynamics after bacterial challenge.*

*However, based on the reviewer's suggestion and for curiosity, we performed the database search and this time included the *Aeromonas* sequences (UniProt Tax_id_380703 downloaded on 2024/06/26) in the search. Nine proteins of *Aeromonas hydrophila* were identified. Of these, only 6 proteins were quantified that too in both Control and AH groups as given in the table below.*

S.no	Protein IDs	Protein name	Peptide counts (unique)	Sequence coverage [%]
1	sp A0KHA2 HCP_AERHH	Hydroxylamine reductase	1	2.4
2	sp A0KQ97 RS7_AERHH	Small ribosomal subunit protein uS7	1	7.7
3	tr A0KHH9 A0KHH9_AERHH	Acetyl-coenzyme A carboxylase carboxyl transferase subunit alpha	1	2.7
4	tr A0KLY9 A0KLY9_AERHH	Outer membrane protein	1	2.5
5	tr A0KQC1 A0KQC1_AERHH	DNA topoisomerase 3	1	2
6	tr A0KQS6 A0KQS6_AERHH	Acetolactate synthase	1	1.6

However, these proteins were identified with low confidence with only one unique peptide and with a sequence coverage from 1.6% to 7.7%. Hence, we didn't include these results in this study.

Also, we would like to confirm that the gut contents were not included in this analysis. The samples were washed in Phosphate Buffered Saline (PBS) 2-3 times to remove any food particles that might be present in the gut tissue. These details have been included in the Methods section of the manuscript as below:

Changes made in the Methods section (under heading *Bacterial challenge and tissue sampling*):

Intestine (midgut and hindgut) samples were collected 48 hours post challenge and washed in Phosphate Buffered Saline (PBS) 2-3 times to remove any remaining food particles that might be present in the gut tissue. This ensures that the samples are clean and not contaminated with external materials that could interfere with subsequent analyses. Collected samples were stored at -80 °C till further use.

Comments from the Editor:

In addition to the comments from Reviewer 1, I would add the following:

1). The description of the tissue used for this study is too vague: was the entire GI (mouth through anus) from each fish used, or one region? How was GI contents removed?

Response: *We appreciate your comment on the description of tissue used. We apologise for not explaining it clearly in the first draft. We would like to state that the intestinal region of the gut (including midgut and hindgut) was collected.*

For cleaning: Before bacterial challenge, fish were starved for 2 days to clear the gut of food residues and undigested food particles. After sample collection, the samples were washed in Phosphate Buffered Saline (PBS) 2-3 times to remove any remaining food particles that might be present in the gut tissue. This ensures that the samples are clean and not contaminated with external materials that could interfere with subsequent analyses.

Considering editor's remarks, we have sufficiently modified this in Methods and throughout the manuscript for better understanding and clarification and the tissue studied and sampling.

2). The title, "Proteomic analysis of gut in *Labeo rohita* reveals ECM as Key Player in host's Response to *Aeromonas hydrophila* Infection" isn't precisely supported by the data from this paper. While the data presented here supports ECM as being involved in Ah infection response, they do not in and of themselves "reveal" this role. Simply relating enriched proteins from this small-scoped study to prior, more rigorous tests made by others who investigated other pathogens and other fish does not necessarily "reveal" that the same phenomena are relevant to the infection system investigated here. Some degree of orthogonal testing, such as the recommendation by Reviewer 1, would better support this notion. Relatedly, strong statements such as, "Such results indicate that integrins and tspan have important role in host response to Ah infection" (page 14) overstate the conclusions that can be drawn from the data the authors generated themselves. Again, linking them to similar prior studies can help them pose new hypotheses, but these links in and of themselves cannot be deemed proof of these hypotheses. Please rephrase this and similar statements throughout the document.

Response: *Thank you for your constructive feedback on our manuscript. We appreciate your careful review and insightful comments. We understand your concern regarding the title and agree that it should more accurately reflect the scope and findings of our study. We have revised the title to better align with the data presented in the manuscript.*

Revised title: *"Proteomic Insights into Extracellular Matrix Dynamics in the Intestine of *Labeo rohita* During *Aeromonas hydrophila* Infection"*

This revised title emphasizes the focus on ECM dynamics and the specific context of the study, avoiding any implications that our work alone "reveals" the role of ECM in the infection response. By using "Insights" instead of "reveals," we believe that the title aligns with the scope of our analysis, which provides valuable observations on ECM dynamics

rather than definitive conclusions. Comments from reviewer 1 has been addressed as above in this document.

Further, as per the suggestion, we have reframed the mentioned and other strong statements throughout the manuscript as the one given below for the mentioned:

Initial statement: “Such results indicate that integrins and tspan have important role in host response to *Ah* infection.”

Revised statement: “These results suggest that integrins and tspan may play a role in the host response to *Ah* infection, consistent with findings from similar prior studies. Further research is needed to confirm their specific functions”.

We hope this addresses your concern, and we look forward to any further feedback you may have.

Re: mSystems00247-24R1 (Proteomic Insights into Extracellular Matrix Dynamics in the Intestine of *Labeo rohita* During *Aeromonas hydrophila* Infection)

Dear Prof. Sanjeeva Srivastava:

Regarding the rigidity experiments suggested by Reviewer 1 and my prior comment about making claims that aren't precisely supported by data, the abstract still reads: "Our findings reveal significant dysregulation in extracellular matrix (ECM) associated proteins during Ah infection, with increased abundance of elastin and Collagen alpha-3(VI) contributing to matrix rigidity." Without measuring matrix rigidity, this statement will need to be scaled back. Alternatively, there are label-free microscopy approaches that could provide the rigidity data you're seeking (reviewed in <https://www.nature.com/articles/s42003-023-04934-8>). From my understanding, at least some would primarily require alternate data analysis methods for interpreting images your existing microscopes can acquire while others may be available through collaboration.

Revision Guidelines

Sincerely,
Joshua Elias
Editor
mSystems

Reviewer #1 (Comments for the Author):

Not sure what to do here.

Response to Reviewers/Editor

Manuscript: mSystems00247-24R1 (Proteomic Insights into Extracellular Matrix Dynamics in the Intestine of *Labeo rohita* During *Aeromonas hydrophila* Infection)

All comments are given due consideration and addressed to the best extent possible. Responses are *italicized in blue*. Text modified in the main document is written in **red**.

=====

Comments from the Editor:

1). Regarding the rigidity experiments suggested by Reviewer 1 and my prior comment about making claims that aren't precisely supported by data, the abstract still reads: "Our findings reveal significant dysregulation in extracellular matrix (ECM) associated proteins during Ah infection, with increased abundance of elastin and Collagen alpha-3(VI) contributing to matrix rigidity." Without measuring matrix rigidity, this statement will need to be scaled back. Alternatively, there are label-free microscopy approaches that could provide the rigidity data you're seeking (reviewed in <https://www.nature.com/articles/s42003-023-04934-8>). From my understanding, at least some would primarily require alternate data analysis methods for interpreting images your existing microscopes can acquire while others may be available through collaboration.

Response: *Thank you for your valuable feedback and the suggestion. We apologize that your previous suggestion was not addressed in our first revision as mentioned for the abstract.*

We acknowledge the concern about the statement: "Our findings reveal significant dysregulation in extracellular matrix (ECM) associated proteins during Ah infection, with increased abundance of elastin and Collagen alpha-3(VI) contributing to matrix rigidity." We agree that, without direct measurements of matrix rigidity, this assertion could be perceived as speculative.

In light of this, we revise the statement in the abstract to more accurately reflect the data we have. The revised sentence reads as "Our findings indicate significant dysregulation in extracellular matrix (ECM) associated proteins during Ah infection, with increased abundance of elastin and Collagen alpha-3(VI)".

This revision ensures that our conclusions remain firmly rooted in the data presented in our study, without making claims beyond our current measurements.

Regarding the suggestion to employ label-free microscopy approaches to directly measure matrix rigidity, we appreciate the reference to the recent review article. While these methodologies indeed offer promising avenues for future research, incorporating such analyses at this stage would require significant additional data collection and collaboration that is beyond our current scope. We are, however, highly interested in exploring these techniques in our future studies to further substantiate and expand upon our current findings.

We hope that this revision addresses the concern satisfactorily and maintains the integrity of our study's findings. Thank you again for your constructive feedback and understanding.

Comments from the reviewer(s):

Reviewer #1 (Comments for the Author):

Not sure what to do here.

Response: *Thank you for your review and for taking the time for our manuscript.*

Re: mSystems00247-24R2 (Proteomic Insights into Extracellular Matrix Dynamics in the Intestine of *Labeo rohita* During *Aeromonas hydrophila* Infection)

Dear Prof. Sanjeeva Srivastava:

Your manuscript has been accepted, and I am forwarding it to the ASM production staff for publication. Your paper will first be checked to make sure all elements meet the technical requirements. ASM staff will contact you if anything needs to be revised before copyediting and production can begin. Otherwise, you will be notified when your proofs are ready to be viewed.

Sincerely,
Joshua Elias
Editor
mSystems